# Development and In Vitro Evaluation of Controlled Release Viagra^®^ Containing Poloxamer-188 Using Gastroplus^™^ PBPK Modeling Software for In Vivo Predictions and Pharmacokinetic Assessments

**DOI:** 10.3390/ph14050479

**Published:** 2021-05-18

**Authors:** Mosab Arafat, Muhammad Sarfraz, Salahdein AbuRuz

**Affiliations:** 1College of Pharmacy, Al Ain University, Al Ain P.O. Box 64141, United Arab Emirates; mosab.arafat@aau.ac.ae (M.A.); Muhammad.sarfraz@aau.ac.ae (M.S.); 2Department of Pharmacology and Therapeutics, College of Medicine and Health Sciences, United Arab Emirates University, Al Ain P.O. Box 17666, United Arab Emirates; 3Department of Biopharmaceutics and Clinical Pharmacy, School of Pharmacy, The University of Jordan, Amman 11942, Jordan

**Keywords:** sildenafil citrate, poloxamer-188, matrix system, polymer, controlled release

## Abstract

Sildenafil is the active substance in Viagra^®^ tablets, which is approved by the FDA to treat sexual dysfunction in men. Poor solubility and short half-life, however, can limit the span of its effectiveness. Therefore, this study focused on an oral controlled release matrix system with the aim to improve solubility, control the drug release, and sustain the duration of drug activity. The controlled release matrices were prepared with poloxamer-188, hydroxypropyl methylcellulose, and magnesium stearate. Various formulations of different ratios were developed, evaluated in vitro, and assessed in silico. Poloxamer-188 appeared to have a remarkable influence on the release profile of sildenafil citrate. In general, the rate of drug release decreased as the amount of polymer was gradually increased in the matrix system, achieving a maximum release period over 12 h. The in silico assessment by using the GastroPlus™ PBPK modeling software predicted a significant variation in C_max,_ t_max_, t_1/2_, and AUC_0-t_ among the formulations. In conclusion, the combination of polymers in matrix systems can have substantial impact on controlling and modifying the drug release pattern.

## 1. Introduction

Sildenafil citrate (SDF) belongs to class II in the biopharmaceutics classification system (BCS) of drugs, having high permeability and low solubility, and being practically insoluble at pH higher than 6 [1]. Since the solubility of SDF considerably varies within the physiological pH ranges of the gastrointestinal tract (GIT), its dissolution and absorption are likely not consistent throughout the GIT depending on the regional environment [2]. SDF is a selective inhibitor of cyclic guanosine monophosphate (cGMP)-specific phosphodiesterase type 5 enzyme (PDE-5), and it is used in the therapy of pulmonary hypertension by selectively inhibiting PDE type 5 pathways in the lungs [3]. It is recommended for selected patients with pulmonary arterial hypertension (PAH), a life-threatening disease, and the drug has favorable effects on endothelial function [4]. SDF was the first oral agent introduced for erectile dysfunction (ED) treatment [3], being rapidly absorbed and effective within 30 min to 1 h. Different clinical trials accomplished efficacious results with single oral doses of SDF between 25 and 100 mg [5]. Overall, medical studies of SDF for over 25 years have confirmed its successful usage [6,7]. In the human body, SDF is *N*-demethylated in liver microsomes, mediated by at least two CYP enzymes, largely metabolized by CYP3A4, but CYP2C9 also exhibited sildenafil *N*-demethylase activity to some extent [3]. 

Only a few studies have been carried out on developing non-oral sustained release SDF dosage forms using polymers, for example, lung-delivered, controlled release formulations of SDF with poloxamer-407 (P-407) [8] or a vaginal controlled release suppository of SDF containing EVAC 210 polymer [9]. In addition, some SDF formulations were prepared with polymers, such as SDF nano-dispersion inhalers incorporating poloxamer-188 (P-188) [10], SDF solid lipid nanoparticles formulated with gelucire 44/14 [11], SDF spray-dried microparticle inhalers formed by using sodium carboxymethylcellulose, sodium alginate, and sodium hyaluronate [12], SDF-loaded polymeric microparticles based on poly(lactide-co-glycolide) [13], and an SDF microemulsion-loaded hydrogel [14]. 

Hydrophilic polymers have been widely applied in drug delivery systems [15] in form of nanoparticles [16,17], microspheres [18], self-emulsifying systems [19], micelles [20,21], cubosomes [22], ophthalmic formulations [23], transdermal delivery [24], nano-solutions [25], and controlled release matrix formulations [26]. For example, several studies have been conducted with the hydrophobic polymer P-188 and/or P-407 to develop stable and controlled release dosage formulations for therapeutic agents, including diclofenac sodium [26], theophylline [27], methotrexate [28], morphine [29], vancomycin [30], tetramethylpyrazine [31], indomethacin [32], ibuprofen [33], and ketorolac tromethamine [34]. To the best of our knowledge, no research has been performed yet on developing a controlled release matrix system for SDF using P-188 or another type of polymer. In this study, a mixture of hydrophilic polymers, namely P-188 and hydroxypropyl methylcellulose (HPMC), combined with hydrophobic release agent magnesium stearate (MGS), was used to form a matrix system to control the release of SDF. First, an oral controlled release matrix system was developed for the drug. Second, in vitro drug release was evaluated and the oral bioavailability and PK parameters of this formulation were assessed using GastroPlus™ Physiological Based Pharmacokinetic (PBPK) modelling 9.8 software. 

## 2. Results

Figure 1 illustrates the release performance of a commercial Viagra^®^ (50 mg) tablet which was used as a reference in this study. Over 90% of the drug was released within 15 min. The result was in good agreement with previously reported data in which 50% of SDF was released in the first 2 min [35]. Measurements were performed in triplicate for the standard calibration curve of SDF, using a UV spectrophotometer at a 292 nm wavelength and distilled water as the dissolution medium. The average regression coefficient r^2^ showed a linear relationship at value of r^2^ = 0.9996.

### 2.1. In Vitro Drug Release 

The profiles in Figure 2 present the SDF release from polymer matrices containing different ratios of P-188 and HPMC to the drug. Gradually increasing the amount of P-188 and HPMC in the matrix system retarded the release rate in the order: Viagra^®^ > F1 > F2 > F3 > F4 > F5 > F6 (polymer: drug). The result shows that the release of SDF can be adjusted in a controlled manner by simply varying the quantity of P-188 and HPMC in the formulation. This phenomenon might be ascribed to the decrease in contact surface area between the drug and dissolution medium with a rising content of P-188 and HPMC. The findings are in good accordance with previous results in the literature [26] where P-188 and HPMC were employed as lipid carriers to slow down the release of diclofenac sodium, a water-soluble drug. It was also observed that the rate of drug release was inversely proportional to the amount of P-188 and HPMC mixed into the matrices.

Figure 3 shows the SDF release profiles of polymer matrices containing different percentages of MGS, incorporated in an increasing order of 3.75, 7.50, and 15.0%. The release rate of SDF from the polymer matrices was further reduced as the quantity of hydrophobic release agent was increased in the preparation. The rate of decrease was in the following order: Viagra^®^ > 3.75% > 7.50% > 15.0% of MGS. Again, the drug release rate could be predictably varied by changing the level of MGS incorporated into the P-188/HPMC matrix system. The decreasing effect of MGS on the release of SDF was due to its hydrophobic character, further reducing the already low hydrophile–lipophile balance (HLB) number of the matrices.

### 2.2. Mechanism of Drug Release from Polymer Matrix

Table 1 summarizes the values for F7, F8, and F9 matrices with corresponding r^2^ values, where r^2^ is the coefficient of determination that indicates how well dissolution data points fit a line. Different given models were applied on the data obtained from the final experimental results. It was observed that the modified release dosage forms of matrices follow zero order release kinetics and the Hixson–Crowell model which describes the erosion of the matrices to release the drug.

### 2.3. In Silico PK of SDF Using PBPK Model Simulations

The physiochemical values in Table 2 were obtained from ADMET Predictor and used to develop and validate a Viagra^®^ (50 mg) tablet (Table 2); some of the physiochemical values were confirmed from the literature [5], and the PBPK model was run for the last three optimum formulations, for example, F_1_, F_2_, and F_3_, as shown in Figure 4. Briefly, the in vitro drug dissolution profiles of each formulation were individually loaded into GastroPlus™ and the simulation was run as a controlled release integral matrix using the PBPK model to predict the plasma concentration time. The resulting profiles versus reference are plotted in Figure 5. Their PK parameters [36], such as C_max,_ t_max_, t_1/2_, AUC_0-∞_ and AUC_0-t_, are outlined in Table 3.

Additional simulations in GastroPlus^TM^ were performed on the in vitro dissolution profiles of F_4_ to F_8_: zero order release kinetics over 24 h (F_4_); release of 35% in 30 min followed by zero order release kinetics to 100% for 24 h (F_5_); release of 70% in 30 min followed by zero order release kinetics to 100% for 24 h (F_6_); sustained release kinetics over 24 h (F_7_); immediate release of 50% followed by a pulsed immediate release of the remaining 50% for 4 h (F_8_). The setup was chosen to see the impact of different dissolution profiles to find the optimum polymer matrix for oral control release of SDF. The plasma concentration time profiles (F_4_–F_8_) versus the reference are presented in Figure 5, and PK parameters [36], like C_max,_ t_max_, t_1/2_, AUC_0-∞_ and AUC_0-t_ are listed in Table 3.

## 3. Discussion

Using P-188 mixed with HPMC clearly showed a delay in SDF release compared to the Viagra^®^ (50 mg) tablet for up to 7 h. Built from ethylene oxide and propylene oxide blocks, P-188 can form an amphipathic structure in which the hydrophobic portion acts as a reservoir for poorly water-soluble drugs [26]. Exposed to the dissolution medium, the polymer slowly hydrates, solubilizing the incorporated drug particles to be finally released into solution.

On the other hand, HPMC itself is able to slow down drug release by forming a gel layer upon swelling when in contact with water [37]. Depending on its concentration in the formulation, the layer can turn from relatively viscous into a thick gel on the surface of the matrix. The quick hydration and gelation of the exterior portion of the matrix system most likely supported the delayed release profiles.

Even though P-188 and HPMC could control the release of SDF, the maximum retardation of 7 h could only be achieved when the ratio of polymers to drug was high. This may not be practical, however, especially for less potent drugs that require high doses, as the dosage form may be not convenient for oral swallowing. Hence, the addition of a hydrophobic excipient in an acceptable amount was required to further extend the release of SDF.

It was observed that 15% MGS enabled the matrix system to release the drug in a controlled manner for up to 12 h. In general, the highly hydrophobic MGS forms a strong surface coating around the other excipients and drug, reducing wettability and water uptake, and slowing down the dissolution rate [38]. Owing to its delaying properties, MGS is a common additive in many solid pharmaceutical formulations, and numerous studies can be found investigating the concentration-dependent effect of MGS on the drug release performance of tablets [39].

Overall, the release of SDF from the polymer matrices was strongly controlled by the dual effects of hydrophilicity and swelling. However, several factors, including physicochemical properties of the drug and excipients, dosage form design, and manufacturing conditions can affect the final release profile [40,41]. Hence, it can help to define the release pattern of the drug as it governs the efficacy of the dosage form. The Hixson–Crowell equation can be applied as a mathematical model for carrier erosion controlled release [26]. Comparing different models, this particular simulation showed the highest correlation coefficient. In conclusion, we can say that drug release of SDF-loaded polymer matrices follows an erosion mechanism.

Furthermore, PBPK modeling studies provide an estimation of PK parameters in order to select the optimum formulation in terms of C_max,_ t_max_, t_1/2_, AUC_0-t_, and oral bioavailability [42,43]. The results can be ultimately used as a prediction of in vivo drug release behavior.

## 4. Material and Methods

### 4.1. Chemicals

Sildenafil citrate (SDF) was purchased from Sigma Aldrich (St. Louis, MO, USA). Poloxamer-188^®^ was obtained from Merck (Darmstadt, Germany), hydroxypropyl methylcellulose (HPMC) was ordered from Thermo-Fisher (Kandel, Germany), and magnesium stearate (MGS) was purchased from Fisher Scientific UK Limited (Loughborough, UK). Viagra^®^ (50 mg) tablets (Pfizer, New York, NY, USA) were obtained from a local retail pharmacy (Abu Dhabi, UAE). All solvents used were of analytical or HPLC grade.

### 4.2. Preparation of Poloxamer 188 Polymer Matrix Containing Sildenafil Citrate

In order to obtain the polymer matrix system, the heat fusion method was applied for series 1. First, P-188 was melted in a water bath at 50 °C and stirred at 250 rpm for 15 min. After the addition of HPMC, stirring was continued at 250 rpm for an additional 15 min. SDF was added, and the mixture was stirred for a further 60 min. The resultant viscous mixture was subsequently poured into size 2 hard gelatin capsules using heated Pasteur pipettes. The capsules were kept upright for approximately 3 to 5 min to allow the matrices to solidify. Afterwards, they were stored in airtight amber bottles containing silica gel at room temperature of 25 °C.

For series 2, SDF-loaded polymer matrices containing a hydrophobic release agent were prepared as follows: P-188 was first melted in a water bath at 50 °C, followed by stirring at 250 rpm for 15 min. HPMC was added, and the mixture continued to be stirred at 250 rpm for 15 min. After the addition of MGS and a further 15 min of stirring, SDF was incorporated with continuous stirring for 60 min. The resultant viscous mixture was subsequently poured into size 00 hard gelatin capsules using heated Pasteur pipettes. The capsules were kept upright for approximately 3 to 5 min to allow the matrices to solidify. Afterwards, they were stored in airtight amber bottles containing silica gel at room temperature of 25 °C.

### 4.3. Preparation of Controlled Release Polymer Matrices Containing Various Drug to Polymer Ratios

Polymer matrix controlled release systems were formed containing different concentrations of the drug (SDF) and ratios of polymers (P-188 and HPMC). In the first part, the ratio of P-188 to HPMC was altered for F1 to F6 in the order: 5:0, 4.5:0.5, 4:1, 3.5:1.5, 3:2, and 2.5:2.5 (*w*/*w*), whereas the drug/polymer ratio was constant at 1:5 (*w*/*w*). The formulations were fabricated following the method described for series 1. The weights of the individual components used are shown in Table 4, and the weight of each capsule was fixed at 275 mg.

In the second part, MGS was incorporated into the polymer matrices, being capable of modifying the rate of drug release. A series of formulations was produced by the addition of an increasing percentage of MGS in the following order: 0, 7.5, and 15%, while the ratio of SDF to polymer mix (P-188/HPMC) was fixed at 1:6:4 (*w*/*w*/*w*), as shown in Table 5. All formulations were prepared as previously described using the method for series 2.

### 4.4. Dissolution Studies

In vitro drug release of SDF from each polymer matrix preparation was determined by dissolution apparatus 2-DT-820 (Erweka GmbH, Heusenstamm, Germany). The pedal method with a 100 rpm rotational speed was used and 900 mL of distilled water served as media at a set temperature of 37 ± 0.5 °C. Samples of 5 mL were withdrawn at the following times: 0.16, 0.33, 0.5, 0.75, 1, 2, 3, 4, 5, 6, 7, and 12 h. All withdrawn samples were filtered and analyzed for drug content using a UV spectrophotometer (detection wavelength was set at 292 nm). The dissolution testing was run for six tablets. In addition, various kinetics models were applied, namely, first order, zero order, Higuchi, and Korsmeyer–Peppas models to find out the release kinetics and mechanism of SDF release. The calibration curve (0.5–64 µg/mL) was prepared with distilled water and analyzed by a UV spectrometer at 292 nm to get the best-fit line and regression equation at value of 0.9995.

### 4.5. Mechanism of Release

To describe the mechanism of drug release from polymer matrix preparations, the dissolution data were analyzed by the Korsmeyer–Peppas mathematical model Equation (1):Q = Kt^n^(1)
where Q is the fraction of drug released at time (t), K is a kinetic release constant incorporating structural and geometrical characteristics of the matrix, and n is the release exponent which indicates the drug release mechanism. Afterwards, the data were fitted to zero order, first order, Higuchi, and Hixson–Crowell models for determining the release kinetics of SDF from the polymer matrices.

### 4.6. Simulation Studies

The PBPK model of a Viagra^®^ (50 mg) tablet was developed in adults using GastroPlus™ (version 9.8. Lancaster, CA, USA). The software is well equipped to integrate the anatomical, physiological, and drug disposition parameters of drugs to build up the PBPK model. Briefly, the method is outlined below: physicochemical and biopharmaceutical parameters were obtained either from the literature or predicted using ADMET Predictor™ 7.2 (Simulation Plus, Inc.), as shown in Table 2. GastroPlus™ 9.8 (Simulation Plus, Inc.) with its Advanced Compartmental Absorption and Transit (ACAT™) model and PBPK Plus™, together with metabolism and transporter modules, were used to build the SDF model for absorption, distribution, metabolism, and elimination. Human organ weights, volume, and blood perfusion rates were generated by the Population Estimate of Age-Related (PEAR™) physiology module in GastroPlus™. The tissue plasma partition coefficients (Kp) were predicted using the default in silico Rodger’s single (Lukacova) Kp method [43]. The metabolic clearance of SDF was estimated from in vitro K_m_ and V_amp_ values of CYP234 and CYP2C9 taken from the literature. The SDF PK parameters (Cl, Vd, compartmental model) were determined by using reported IV data [5] and built in the PK plus tool in GastroPlus^TM^ software. The PBPK model was validated using literature data for healthy volunteers [5]. The mechanistic data provided by the PBPK model were applied to investigate the PK profile of in vitro SDF controlled release from the matrix system.

### 4.7. Statistical Analysis

A statistical analysis was performed using one-way analysis of variance to compare the mean value of each variable, considering a result statistically significant when *p* ≤ 0.05.

## 5. Conclusions

Controlled release polymer matrices of SDF were prepared by incorporating the drug into a mixture of P-188, HPMC, and MGS at various ratios. The combination of P-188 and HPMC was able to extend the drug release, however, a high proportion of polymer mixture (≥90%) was required. Replacing a portion with MGS obviously helped to slow down the drug release for up to 12 h. The PK parameters of the matrix systems with SDF significantly differed from a Viagra^®^ (50 mg) tablet in the extent and rate of absorption and bioavailability. Therefore, the in silico model assisted in tailoring the optimum PK shape of the controlled release preparation. Controlled release of SDF might not only help to avoid the risk of side effects caused by burst release, but could also make the drug more accessible for men with pre-existing conditions, supporting their psyche, or enhancing their overall life quality. In addition, the concept can be transferred to other treatments such as for pulmonary hypertension, as the typical treatment is either IV or several oral doses per day.

## Figures and Tables

**Figure 1 pharmaceuticals-14-00479-f001:**
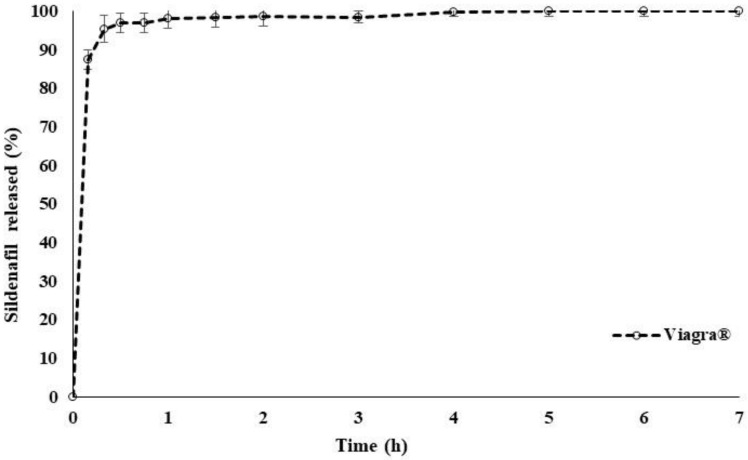
SDF release profile from branded drug Viagra^®^ 50 mg in distilled water at 37 °C (*n* = 6).

**Figure 2 pharmaceuticals-14-00479-f002:**
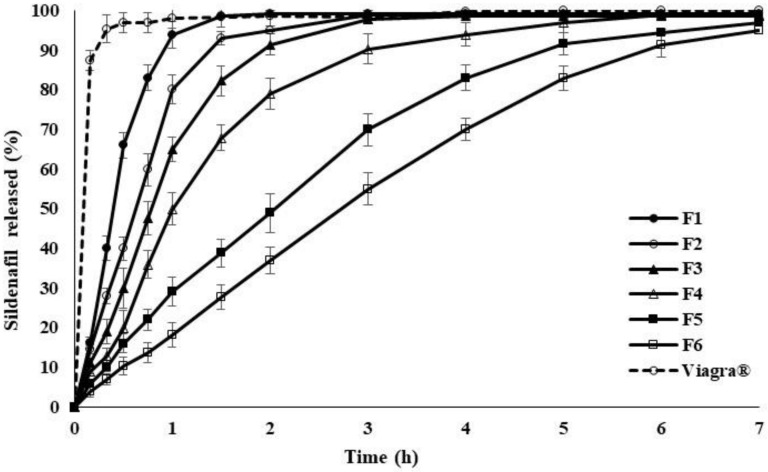
The percentage of SDF release from polymer matrices of various ratios of P-188 and HPMC over 7 h, using distilled water at 37 °C, values are means ± S.D. (*n* = 6).

**Figure 3 pharmaceuticals-14-00479-f003:**
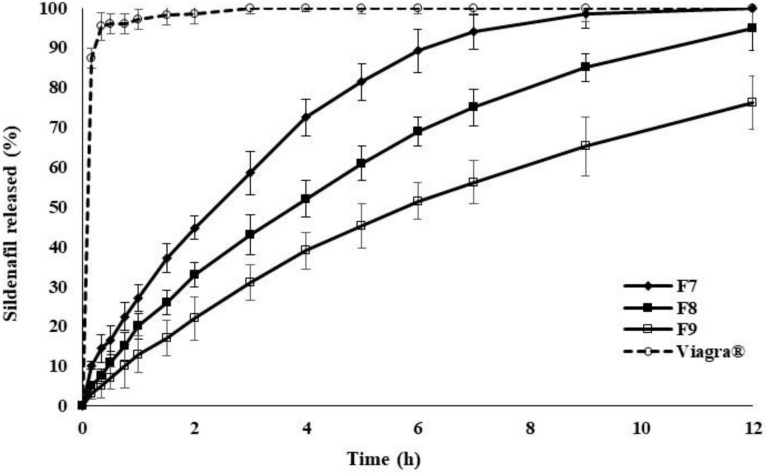
The percentage of SDF release from polymer matrices containing various drug to polymer ratios with the addition of MGS and STA over 12 h, using distilled water at 37 °C, values are means ± S.D. (*n* = 6).

**Figure 4 pharmaceuticals-14-00479-f004:**
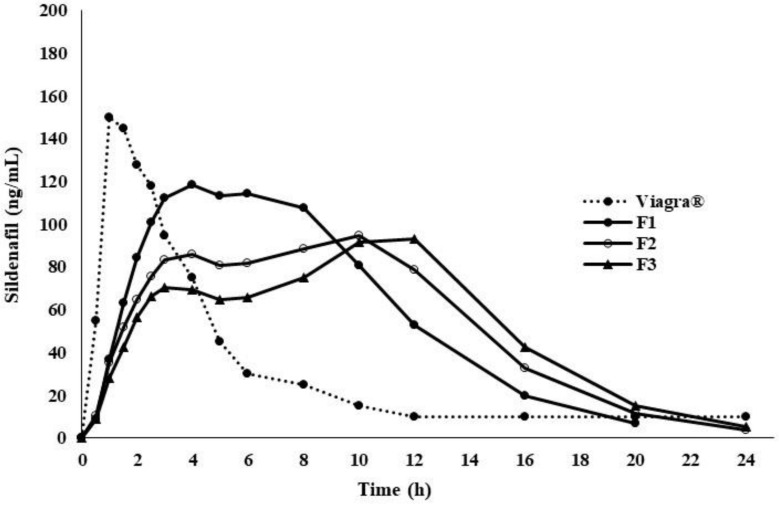
Simulated plasma concentration–time profiles of sildenafil oral controlled release polymer matrices formulation (F1–F3) versus Viagra^®^ 50 mg using Gastroplus™ PBPK modeling software.

**Figure 5 pharmaceuticals-14-00479-f005:**
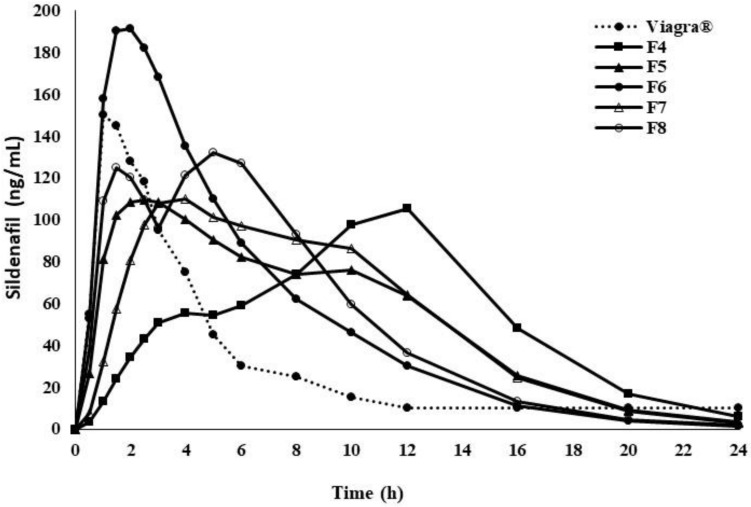
Simulated plasma concentration–time profiles of sildenafil oral controlled release polymer matrices formulation (F4–F8) versus Viagra^®^ 50 mg using Gastroplus™ PBPK modeling software.

**Table 1 pharmaceuticals-14-00479-t001:** Mechanism of SDF release from controlled release polymer matrices using various kinetic release models represented by r^2^.

Model Name	r^2^ of F7	r^2^ of F8	r^2^ of F9
Zero Order Model	0.9883	0.9825	0.9754
First Order Model	0.6394	0.5738	0.5769
Hixson–Crowell Model	0.9850	0.9907	0.9875
Higuchi Model	0.9601	0.9685	0.9696
Korsmeyer–Peppas Model	0.4203	0.3309	0.3290

**Table 2 pharmaceuticals-14-00479-t002:** Physicochemical and biopharmaceutical values of Viagra^®^ 50 mg obtained from ADMET Predictor in GastroPlus™ software (version 9.8, Simulation Plus, Inc., Lancaster, CA, USA).

Parameter	Values
**Log P**	^a^ 2.27
**Molecular Weight (g/mol)**	^a^ 474.59
**Ionization Constant**	(pKa) 6.5
**Solubility (mg/mL)**	^a^ 6.965 (pH = 3)
**Diffusion Coefficient (cm^2^ /sec × 10^−5^)**	^a^ 0.58 × 10^−5^
**Papp (Caco-2)**	^a^ 7.077 (pH = 4)
**Jejunal Effective Permeability (Peff) (×10^−4^ cm/s)**	^a^ 3.48 × 10^−4^
**Unbound Percent in Human Plasma (Fup %)**	^a^ 28.10
**Human Blood-to-Plasma Concentration Ratio (Rbp)**	1.65

^a^ is referring to using ADMET predictor in GastroPlus™ (version 9.8, Simulation Plus, Inc., Lancaster, CA, USA).

**Table 3 pharmaceuticals-14-00479-t003:** Pharmacokinetic parameters of SDF after in silico oral administration of SDF in controlled release polymer matrices containing various drug to polymer ratios. Data obtained after PBPK modeling of each formulation.

PK	Viagra^®^	F1	F2	F3	F4	F5	F6	F7	F8
**C_max_ (ng mL^−1^)**	159	120.9 *	96.2 *	93.9 *	105.1 *	109.3 *	193.2 *	113.7 *	132.4 *
**t_max_ (h)**	1.46	3.68 *	9.44 *	11.2 *	12.08 *	2.48 *	1.76	3.61 *	5.28 *
**t_1/2_ (h)**	4.07	5.65 *	8.12 *	9.85 *	11.8 *	7.75 *	6.10 *	8.12 *	6.51 *
**AUC_0-∞_** **(ng mL^−1^h)**	530	128 *	1267 *	1258 *	1252.2 *	1272.2 *	1288.7 *	1237.7 *	1289.9 *
**AUC_0-t_ (ng mL^−1^h)**	528	1271 *	1252 *	1238 *	1224.6 *	1259.5 *	1283.2 *	1262.1 *	1276.5 *

* *p < 0.05* vs. Viagra^®^ 50 mg.

**Table 4 pharmaceuticals-14-00479-t004:** Controlled release polymer matrices containing various drug to polymer ratios.

Formulations	SDF (mg)	P-188 (mg)	HPMC (mg)	Total Weight (mg)	Ratio 1:5
F1	50	250	0	275	1:5:0
F2	50	225	25	275	1:4.5:0.5
F3	50	200	50	275	1:4:1
F4	50	175	75	275	1:3.5:1.5
F5	50	150	100	275	1:3:2
F6	50	125	125	275	1:2.5:2.5

**Table 5 pharmaceuticals-14-00479-t005:** Controlled release polymer matrices containing various drug to polymer ratios with the addition of MGS and STA.

Formulations	SDF (mg)	P-188 (mg)	HPMC (mg)	STA (mg)	MGS (mg)
F7	50	300	200	50	0
F8	50	300	200	50	37.5
F9	50	300	200	50	75

## Data Availability

The data presented in this study are available on request from the corresponding author. The data are not publicly available due to privacy or ethical restrictions.

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
