# Peer review of "Development and In Vitro Evaluation of Controlled Release Viagra^®^ Containing Poloxamer-188 Using Gastroplus^™^ PBPK Modeling Software for In Vivo Predictions and Pharmacokinetic Assessments"

_pharmaceuticals, 2021, doi:10.3390/ph14050479_

Round 1
Reviewer 1 Report
The manuscript “DEVELOPMENT AND IN VITRO EVALUTION OF CONTROLLED RELEASE VIAGRA® CONTONING POLOXAMER-188 USING GASTROPLUS™ PBPK MODELING SOFTWARE FOR IN VIVO PREDICTIONS AND PHARMACOKINETIC ASSESSMENTS” need extensive English correction many mistakes can be found throughout the text, the first one here in the title (contoning). Missing letters, parentheses or commas can be found frequently. A thorough correction regarding this is mandatory.
The preparation of controlled release drug delivery systems based on polymers is interesting and as such a good idea for many drugs. For Sildenafil used in this work the benefit is not completely clear to me. As a starting point I’m not sure if this formulation should be used as a treatment for erectile disfunction or pulmonary hypertension. Both are mentioned in the manuscript and I was not able to find an explanation if there are different plasma concentration curves needed for these very different treatments and for which treatment the dosage forms are planned.
In figure 5 the simulated plasma concentration-time profiles of the formulations 5-8 do not fit to the release curves in figure 2 and 3. Formulation 5 and 6 revealed slower release than formulation 4 and the plasma concentration-time profiles are depicted with a higher and/or earlier cmax. Such a profile should not be possible after a slower release of the drug.
In the discussion (line 193) the authors state that MGS melts during the particle preparation in the water bath and coat the drug particles with molten MGS. The melting point of MGS is at approx. 88°C, please explain how you were able to reach this in a water bath thermostated to 50°C.
In line 256 the authors state that the ratio of drug to polymer is 1:1 (w/w) in Table 4 it is clearly stated that 50mg sildenafil were mixed with a total amount of 250mg polymer, that is not a 1:1 ration.
In line 280 the calibration curve of sildenafil was prepared with similar solvent, please give more detailed information on the solvent similar to distilled water you used here.
In line 298 and 299 Table 4 is cited to contain information on physicochemical and biopharmaceutical parameters. Table 4 in my version of the manuscript contains information in matrix compositions not biopharmaceutical parameters, they are given in table 2. In this table 2 (line 139) there is a small a in every line to which I can’t find any explanation anywhere in the manuscript. In the text it is mentioned that the parameters were obtained from the literature but no reference is mentioned. Include all references used please.
Author Response
SUMMARY OF CHANGES
Dear Reviewer, 1
Thank you for your valuable comments on this manuscript. All raised up comments have been responded below and all required modifications have been done on this manuscript accordingly.
Responding to all points by reviewer.
|
1 |
|
|
2 |
|
|
3 |
|
|
4 |
|
|
5 |
· Comment 5: In line 256 the authors state that the ratio of drug to polymer is 1:1 (w/w) in Table 4 it is clearly stated that 50mg sildenafil were mixed with a total amount of 250mg polymer, that is not a 1:1 ration.
|
|
6 |
· Reopens 6: Yes, it has been fixed to distilled water. (highlighted in red color, lines 247) |
|
7 |
· Comment 7: In line 298 and 299 Table 4 is cited to contain information on physicochemical and biopharmaceutical parameters. Table 4 in my version of the manuscript contains information in matrix compositions not biopharmaceutical parameters, they are given in table 2. In this table 2 (line 139) there is a small a in every line to which I can’t find any explanation anywhere in the manuscript. In the text it is mentioned that the parameters were obtained from the literature but no reference is mentioned. Include all references used please
|

Reviewer 2 Report
Dear Authors,
Thank you for submitting valuable manuscript to our journal.
The paper describes the development of controlled release form of Viagra tablet using prediction software.
In the point of view as a reader, I am curious of the needs to develop the controlled release tablet of Viagra since this goals fast action of the moment. If the market needs arise, the manufactures may already release controlled release type of sildenafil but still it is not showed in the market. That's why the novelty of this research grades lower.
There are minor points to be revised;
- Titles, subtitles were to be unified (some has numbers, others not)
- Page 2, reference No. 18 should have bracket.
- Page 6, legend of Table 3 should be attached with table
- Page 7, line 153 Gastroplus TM. (space) The~
- Page 7, line 154 as follow:
- Page 7, line 171 abstaining?
- Page 7, line 185 may be not con(should be removed) convenient
- Page 8, degrees Celcius needs to be selected right one
- Page 9, line 254 series (space) 1
- Page 9, line 273-274: the sentence needs to be revised
- Page 10, line 287 equation font size and style need to be revised
- Page 11 reference font size and style need to be unified
One thing I suggest is, if the material is something that which targets other diseases, (ex. hypertension, diabetes etc) need to act longer time, this approach may fit the needs in the market. But sildenafil is not used for controlled release but short time acting.
Thanks
Author Response
SUMMARY OF CHANGES
Dear Reviewer, 2
Thank you for your valuable comments on this manuscript. All raised up comments have been responded below and all required modifications have been done on this manuscript accordingly.
Responding to reviewer curious question.
Question1: I am curious of the needs to develop the controlled release tablet of Viagra since this goals fast action of the moment. If the market needs arise, the manufactures may already release controlled release type of sildenafil but still it is not showed in the market.
Answer 1: Thanks for this valid point. We like to highlight three reasons why a slow release can be important: (1) Sildenafil is also used in other treatment; (2) With no burst release it can lower the risk of side effects such as arrhythmia and might make it accessible for men with pre-existing conditions like high blood pressure; (3) With the right balance of drug content and effectiveness a slow release can help to release pressure on men with erectile dysfunction and enhance their psyche and overall life quality.
Responding to all minor points by reviewer.
|
1 |
Comment 1: Titles, subtitles were to be unified (some has numbers, others not) Reopens 1: Yes, it has been fixed and highlighted in red color in the entire manuscript. |
|
2 |
Comment 2: Page 2, reference No. 18 should have bracket Reopens 2: Yes, it has been fixed and highlighted in red (line 60). |
|
3 |
Comment 3: Page 6, legend of Table 3 should be attached with table Reopens 3: Yes, it has been attached and highlighted in red color (line 145-147). |
|
4 |
Comment 4: Page 7, line 153 Gastroplus TM. (space) The~ Reopens 4: Yes, it has been fixed and highlighted in red color (line 148). |
|
5 |
Comment 5: Page 7, line 154 as follow: Reopens 5: Yes, it has been fixed and removed from the text (highlighted in red color, line 150). |
|
6 |
Comment 6: Page 7, line 171 abstaining? Reopens 6: Yes, it has been corrected and the entire sentence has been paraphrased to “Exposed to the dissolution medium, the polymer slowly hydrates, solubilizing the incorporated drug particles to be finally released into solution” (line 164-168) |
|
7 |
Comment 7: Page 7, line 185 may be not con(should be removed) convenient Reopens 7: Yes, it has been fixed to “as the dosage form may be not convenient for oral swallowing” (highlighted in red color, line 172) |
|
8 |
Comment 8: Page 8, degrees Celcius needs to be selected right one Reopens 8: Yes, it has been fixed and corrected (highlighted in red color, line 212, 214 and 220) |
|
9 |
Comment 9: Page 9, line 254 series (space) 1 Reopens 9: Yes, it has been fixed (highlighted in red color, line 226) |
|
10 |
Comment 10: Page 9, line 273-274: the sentence needs to be revised Reopens 10: Yes, it has been fixed paraphrased to “Pedal method with 100 rpm rotational speed was used and 900 mL of distilled water served as media at a set temperature of 37 ± 0.5 °C. Samples of 5 mL were withdrawn at the following times: 0.16, 0.33, 0.5, 0.75, 1, 2, 3, 4, 5, 6, 7, and 12 h. All withdrawn samples were filtered and analyze for drug content using UV-spectrophotometer (wavelength was set at 292 nm)”, (highlighted in red color, line 239-244) |
|
11 |
Comment 11: Page 10, line 287 equation font size and style need to be revised Reopens 11: Yes, it has been fixed (highlighted in red color, line 252) |
|
12 |
Comment 12: Page 11 reference font size and style need to be unified Reopens 12: Yes, it has been fixed (highlighted in red color, lines 307-409). |
Responding to reviewer suggestion.
Suggestion 1; if the material is something that which targets other diseases, (ex. hypertension, diabetes etc) need to act longer time, this approach may fit the needs in the market. But sildenafil is not used for controlled release but short time acting.
Reopens 1: Thank you for this valuable suggestion. Yes, this suggestion will be considered for next research study aiming on formulation of controlled release antihypertensive drug using a mixture of polymers containing P-188.

Round 2
Reviewer 2 Report
Dear Authors,
Thank you for all the detailed answers.
I think almost of uncertainties are resolved.
Thanks